# Aetiologies and Risk Factors of Prolonged Fever Admission in Samtse Hospital, Bhutan, 2020

**DOI:** 10.3390/ijerph19137859

**Published:** 2022-06-27

**Authors:** Tsheten Tsheten, Karma Lhendup, Thinley Dorji, Kinley Wangdi

**Affiliations:** 1Department of Global Health, National Centre for Epidemiology and Population Health, College of Health Medicine, Australian National University, Acton, Canberra, ACT 2601, Australia; tsheten.tsheten@anu.edu.au; 2Samtse Hospital, Samtse 22002, Bhutan; karyang4@gmail.com; 3Jigme Dorji Wangchuck National Referral Hospital, Thimphu 11001, Bhutan; dorji.thinleydr@gmail.com; 4Kidu Mobile Medical Unit, His Majesty’s People’s Project, Thimphu 11001, Bhutan

**Keywords:** developing countries, prolonged fever admission, infections, health services, epidemiology, aetiologies, risk factors

## Abstract

Febrile illness is a common cause of hospital admission in developing countries, including Bhutan. Prolonged fever admission can add considerable strain on healthcare service delivery. Therefore, identifying the underlying cause of prolonged hospital stays can improve the quality of patient care by providing appropriate empirical treatment. Thus, the study’s aims were to evaluate the aetiologies and factors of prolonged fever admission in Samtse Hospital, Bhutan. Fever admission data from 1 January to 31 December 2020 were retrieved from the Samtse Hospital database. Prolonged hospital stay was defined as those with >5 days of hospital admission. Univariable and multivariable logistic regression was used to identify risk factors for a prolonged hospital stay. Of 290 records, 135 (46.6%) were children (≤12 years), 167 (57.6%) were males, and 237 (81.7%) patients were from rural areas. The common aetiologies for fever admissions were respiratory tract infection (85, 29.3%) and acute undifferentiated febrile illness (48, 16.6%). The prolonged stay was reported in 87 (30.0%) patients. Patients from rural areas (adjusted odds ratio [AOR] = 4.02, 95% CI = 1.58–10.24) and those with respiratory tract infections (AOR = 5.30, 95% CI = 1.11–25.39) and urinary tract infections and kidney disease (AOR = 8.16, 95% CI = 1.33–49.96) had higher odds of prolonged hospital stay. This epidemiological knowledge on prolonged hospital stay can be used by the physician for the management of fever admission in Samtse Hospital.

## 1. Introduction

Fever is an important feature of bacterial, viral, parasitic, and fungal infections [1]. Viral infections such as dengue, influenza and yellow fever, and bacterial infections such as leptospirosis, typhoid and scrub typhus account for more than 80% of the pathogens causing febrile illness [2,3]. In South-East Asia and Latin America, dengue is the leading cause of infectious disease associated with fever [2,3,4]. Fever can also occur as a response to non-infectious aetiologies [5], such as trauma, autoimmune diseases, inflammatory conditions, environmental stressors, and drugs [6]. Neurological disorders such as stroke and brain injuries also present with fever and are often associated with the worst health outcomes [7].

Fever and febrile response contribute to pathogenesis, clinical presentation, and outcome of many diseases and disorders. The majority present with malaise and fatigue and may not be associated with any localising symptoms of infection [8,9]. Many preventable deaths occur in low- and middle-income countries because of incorrect or delayed diagnosis, largely due to limited diagnostic and treatment facilities [3]. Cases of fever for prolonged duration without a known aetiology present a challenge for both the physicians and the patients, often leading to prolonged length of hospitalisation [10]. Prolonged hospital stay is associated with poor health outcomes, increased risk of mortality [11], substantial economic burden and a huge psychological impact on the patients and their families [12,13].

Bhutan is a middle-income country located in the eastern Himalayan Mountain range in South Asia, where the burden of fever admissions has not yet been studied. In the southern districts of Bhutan, increasing incidence of dengue and scrub typhus are responsible for increasing hospital visits and admissions [14,15,16,17]. Other reported causes of fever admissions include typhoid, chikungunya and respiratory illness [18,19], while the incidence of malaria has been on the decline in recent years [16,20]. In Bhutan’s efforts to provide free and quality healthcare to all, there is a need for a better understanding of the epidemiological characteristics and aetiologies of fever and the length of hospital stay among the population living in the sub-tropical region in Bhutan. Results from such studies can help improve the quality of patient care by providing appropriate empirical treatment. This study investigated the aetiologies and risk factors of prolonged fever, defined as that lasting >5 days of admission due to fever in Samtse Hospital, Bhutan.

## 2. Materials and Methods

### 2.1. Study Design

This was a retrospective study using the hospital records of fever admissions.

### 2.2. Study Setting

Healthcare in Bhutan is provided through a three-tiered system: primary health centre (PHC), sub-post and outreach clinics at the primary level; district and general hospitals at the secondary level; and referral hospitals with specialist services at the tertiary level. The three referral centres are located in geographically strategic locations in the west (Thimphu), east (Monggar), and central (Gelephu) regions.

Samtse District is located in the sub-tropical region of the southern foothills of Bhutan [21]. It has a 40-bedded and three 10-bedded hospitals, 12 PHC, and 5 sub-posts (Figure 1). Samtse Hospital (40 beds) is located a few kilometres away from the international border with India. The hospital is operated by eight doctors (four general doctors and four specialists), 28 nurses and 16 supporting staff. It provides care to the patients referred from other health facilities within the district and often from other neighbouring districts such as Chukha. In 2020, the hospital provided care to 51,951 outpatient and 2186 inpatient cases. Patients’ background in Samtse Hospital is not different from other district hospitals in the country.

The hospital has facilities for routine blood investigations, rapid diagnostics tests (for malaria, dengue, scrub typhus), X-ray and ultrasonography, Gram staining and routine microscopic examinations, and GeneXpert test for tuberculosis. The clinical samples (blood, stool, sputum, nasal/throat swabs) for additional tests, such as influenza, tuberculosis, HIV, rotavirus, measles and rubella, are sent to the Royal Centre for Disease Control (RCDC) in Thimphu (capital) on weekly basis via Bhutan Post.

Fever cases are admitted to the inpatient department in three ways: (i) Referrals from PHC, (ii) From among outpatients with fever-related illness, and (iii) Emergency referrals or emergency admission. There are separate wards for males, females, maternity, paediatrics and surgical cases.

### 2.3. Study Population

All patients admitted with fever-related complaints between 1 January and 31 December 2020 were retrieved. Inclusion criteria were: patients of any age and sex, having fever as determined and recorded by the treating physicians and records containing diagnosis, clinical signs, and symptoms. Exclusion criteria were: records with no laboratory test reports and incomplete information on patient demographic/clinical characteristics.

### 2.4. Data Collection

Data were extracted from the admission registers and patient files maintained by the indoor department of the hospital. The structured pro forma collected information on socio-demographic characteristics (age, gender, occupation, residential location), clinical profile (diagnosis, symptoms, duration of admission), seasonality (monthly trend) and outcome of the patient (duration of admission and recovery status) (Appendix A).

### 2.5. Case Definition

Fever was defined as at least one episode of axillary temperature ≥ 98.9 °F in the morning or ≥99.9 °F in the evening at the time of admission recorded through a calibrated thermometer. Length of stay was considered from the day of admission to the hospital to the day of discharge or death; prolonged hospital stay was that lasting for >5 days of hospital admission, and short stay was that lasting ≤5 days [22,23]. Aetiologies of fever were recorded as determined clinically and noted in the clinical charts. For this study, aetiologies were categorised as: acute gastroenteritis, respiratory tract infection (bronchiolitis, pleural effusion, tuberculosis, lower respiratory tract infection and pneumonia), fever of unknown origin or acute undifferentiated febrile illness (AUFI-refers to other febrile illness not confirmed with the specific laboratory test), sepsis, scrub typhus and others. Here, others comprised a list of other aetiologies combined due to their low number of cases and included abdominal distension, candidiasis, dengue fever, malaria, rabies, tonsillitis, parotitis, neurological disease, typhoid fever, abscess, hepatitis and appendicitis.

### 2.6. Statistical Analysis

Data were entered into MS Excel 2016 (Microsoft Cooperation), double-checked, and validated for accuracy. Statistical analysis was conducted using STATA version 16 (Stata Corporation, College Station, TX, USA), and ArcGIS (ESRI, Redlands, CA, USA) was used to create the map. Data are described using frequencies and proportions for categorical variables, while the median and interquartile range were used for the continuous data. Pearson’s chi-squared test or Fisher’s exact test was used to compare the proportion between different categorical variables. For continuous variables, non-parametric tests, Mann–Whitney U test, and Kruskal–Wallis test was used to compare the differences between two and three groups, respectively. Univariable and multivariable logistic regression models were used to identify risk factors for a prolonged hospital stay. Any variable with a *p*-value < 0.2 in the univariable analysis was considered a candidate variable in the multivariable model [24]. All potential independent variables were entered in the full model, and associations were reported using adjusted odds ratios (AOR) with 95% confidence intervals (CI). In the final model, those variables with *p*-value < 0.05 was considered statistically significant.

## 3. Results

### 3.1. General Characteristics

There were 297 patients admitted to the hospital in 2020 with fever (Figure 2). Of those, seven records did not have complete data and were excluded from the analysis. Of 290 records included in the study, 135 (46.6%) were children aged ≤ 12 years. The majority of the patients were males (167, 57.6%) and lived in rural areas (237, 81.7%). Those who recovered and discharged predominated the fever admissions (260, 90.0%), 25 (8.7%) were referred for specialised care to the National Referral Hospital, Thimphu), and 4 (1.4%) died (Table 1).

The overall median duration of hospital stay was four (interquartile range (IQR): 3–7) days. There were 87 (30.0%) patients who had a prolonged stay (Figure 2). The median duration of hospital stay was eight (IQR: 7–10) days for those with prolonged stays, while the median of hospital stay was three (IQR: 3–4) days for those with a short stay. The median length of stay for each demographic variable is presented in Appendix A.

The proportions of children and adults with prolonged stay were statistically not significant as compared to those with short stay (*p* > 0.05). Males had a higher proportion of prolonged stay than females; however, it was statistically not significant. According to the occupational groups, children/students had the highest proportion, followed by farmers and housewives. The least observed occupation group was others that comprised of civil servants, armed personnel, and monks. Patients residing in rural settings were more frequently observed to have prolonged admission than those residing in urban areas, and the difference was statistically significant (*p* = 0.002) (Table 1).

### 3.2. Clinical Characteristics

One-fifth (56, 19.3%) of patients had co-morbidities, mostly related to hypertension (12, 21.4%), diabetes (7, 12.5%) and alcohol liver disease (7, 12.5%). The other forms of co-morbidities were anaemia, heart disease, cancer, seizure, malnutrition, stroke and sexually transmitted infection. No significant difference was found between co-morbidities and the length of stay (Table 1).

Cough (121, 41.3%) was the most common symptom followed by difficulty in breathing (67, 22.9%), vomiting (38, 13.0%), abdominal pain (31, 10.6%), diarrhea (30, 10.2%), and dysuria (25, 8.5%). Symptoms accompanying fever were identified and presented according to the length of stay in Figure 3.

### 3.3. Temporal Pattern

Higher and distinct peaks of escalated fever cases were observed between January and March, and another smaller but distinct peak was observed between July and September. However, a much longer duration, for example, about five years, might be required to substantiate the temporal pattern (Figure 4).

### 3.4. Aetiologies of Fever Admission

Respiratory infection was the most common cause of fever admission (85, 29.3%), predominantly associated with the infection of the lower respiratory tract. The other common causes of admission were acute undifferentiated febrile illness (8, 16.6%), sepsis (and septic shock) (44, 15.2%), acute gastroenteritis (20, 6.9%), urinary tract infection and kidney disease (17, 5.9%), and scrub typhus (10, 3.5%). A significant difference was found between these aetiologies and prolonged hospitalisation (*p* = 0.016) (Table 2).

### 3.5. Factors Associated with Prolonged Length of Stay

In the univariable analysis, the patient’s residence, respiratory infection, urinary tract infection and kidney diseases were significantly associated with the prolonged stay. In the multivariable logistic regression, after adjusting for other variables, patients from rural areas were four times (AOR = 4.02, 95% CI = 1.58–10.24) more likely to have prolonged stay compared to those living in urban areas. Co-morbidities such as respiratory tract infection, urinary tract infection and kidney diseases were five (AOR = 5.30, 95% CI = 1.11–25.39) and eight times (AOR = 8.16, 95% CI = 1.33–49.96) at odds of prolonged stay compared to acute gastroenteritis. Other socio-demographic characteristics, including age, sex and occupation, were not statistically significant (Table 3). Further analysis with cross-tabulation found a significant difference between patients living in rural and urban areas in relation to co-morbidities; 52 patients living in rural areas were found to have one form of co-morbidities, whereas only four patients living in urban areas had the co-morbidities (Appendix A).

## 4. Discussion

This is the first study to describe the length of hospitalisation and clinically-determined aetiologies of fever admissions in Samtse Hospital, Bhutan. The median length of hospital stay for fever admission was four days, and the proportion of patients with prolonged hospitalisation (>5 days) was 30.0%. Respiratory infection and AUFI were the most common cause of fever admissions. Patients living in rural areas, respiratory infection, urinary tract infection and kidney disease were positively associated with the prolonged hospitalisation with febrile illness.

In the present study, the prolonged hospital stay was higher among people living in rural areas than those living in urban areas. Other studies have shown people living in rural areas are more prone to infections, adverse health outcomes and higher admission rates due to a lack of awareness of disease prevention [25], poverty, poor socioeconomic and poor dietary habits [26]. This could be true in Bhutan because the rural population has fewer resources to obtain health education. Due to a lack of adequate health literacy, patients might seek treatment at the advanced stage of disease or in serious conditions. In addition, patients from rural areas are more likely to be undernutrition because most rural people practice subsistence farming and have limited access to nutritious and wholesome food.

Respiratory tract infection is the most common reason to seek medical care and hospital admission and is a leading cause of childhood mortality in developing countries [27]. Although the notification of respiratory tract infection decreased in Bhutan due to COVID-19 restrictions, it is still a leading cause of infectious disease in the country [28]. The first case of COVID-19 in the Samtse District was reported on 9 May 2021 [29]. There are a number of plausible reasons for this finding. First, the traditional method of cooking in rural Bhutan is firewood, and this can contribute to higher respiratory illness [30,31]. Secondly, air pollution in southern Bhutan, where Samtse is located, is expected to be higher with increased risk [32,33,34,35]. An earlier study showed that pneumonia was significantly higher in some parts of the southern sub-districts in Bhutan [19]. Lastly, the surface water from streams is the main source of drinking water in rural Bhutan [36]. Unsafe drinking water and sanitation are important drivers of respiratory illness, including pneumonia [37]. Encouraging people to improve drinking water through boiling and filtration should be encouraged to reduce respiratory illness in Samtse and Bhutan.

The next common cause of prolonged stay in the present study was AUFI, although it was not statistically significant but worth discussing here. AUFI continues to pose a serious clinical challenge with an increasing frequency of undiagnosed cases. This impedes physicians’ decisions in the clinical judgement of accurate treatment, although many cases might be self-limiting viral fevers. A recent review found that as much as 16–55% of the infections are reported as AUFI [38]. In the present study, about 16.6% of the febrile illnesses were diagnosed as AUFI. AUFI is diagnosed based on the history and physical examination of the patient without identifying the etiological agent [38]. Samtse Hospital lacks adequate laboratory capacity to identify important bacterial and viral pathogens that may cause AUFI. For example, the herpes virus has been reported to cause ~40% of all AUFI; however, the capacity to identify this virus is not available even at the national reference laboratory in Bhutan [39]. Reliable diagnostic tools are required to make appropriate treatment decisions.

Kidney disease is a critical condition that complicates the clinical course of a patient and is linked with higher mortality and a longer duration of hospital stay [40]. In a study involving a cohort of 2004 patients in the United States, kidney disease has been identified as the predictor of a prolonged hospital stay along with other risk factors, such as increasing age and female sex [41]. A large study involving 40,869 also reported the same finding that demonstrated a significant association between prolonged length of stay and renal disease [42]. In another study, febrile infants with urinary tract infections were hospitalized more than the average length of stay to receive intravenous therapy and for better care [43]. Discordant antibiotic therapy (in vitro non-susceptibility of the uro-pathogen to initial antibiotic) is common and associated with a longer duration of hospitalization [44]. The association identified between prolonged stay and urinary tract infection and kidney disease in the current study is consistent with these previous studies.

Our study has several limitations. First, the cut-off value of prolonged hospital stay is based on previous studies. This cut-off value is still subjective and disputed [22]. Second, the major limitation of this study is the use of hospital admission data. The completeness and representativeness of such data cannot be ascertained. Third, this is a retrospective single-centre study, and findings might not be generalised to the whole country. Further, other unmeasured risk modifiers such as socioeconomic development, living standards, and treatment-seeking behaviour were not included in this study because these were not in the patient records. Therefore, a prospective study with additional variables in all other hospitals in the southern districts might provide additional information that can be generalizable to the whole country. Finally, the inclusion of causal organisms would have provided a better understanding of prolonged admission, but blood culture and similar parameters are not routinely performed in most district hospitals in Bhutan. Notwithstanding these limitations, this is the first study in Bhutan to identify the risk factors and epidemiology of prolonged fever admissions. The findings provided invaluable evidence of prolonged fever admission to Samtse Hospital.

## 5. Conclusions

Respiratory tract infection and AUFI were the most common causes of fever admission in 2020. Prolonged hospital stay was commonly observed among patients from rural areas and with co-morbidities such as respiratory infections, urinary tract infections, and kidney diseases. The physicians and health professionals of Samtse Hospital should be informed of this epidemiological knowledge of prolonged fever admission for better fever management. The laboratory capabilities in Samtse Hospital should be strengthened to support physicians in the management of fever admission, particularly AUFI. A nationwide study on prolonged fever admission would be useful in developing evidence-based diagnostic criteria and treatment guidelines. Further, it is suggested to compare prolonged stay due to fever and non-fever admissions to provide effective patient care.

## Figures and Tables

**Figure 1 ijerph-19-07859-f001:**
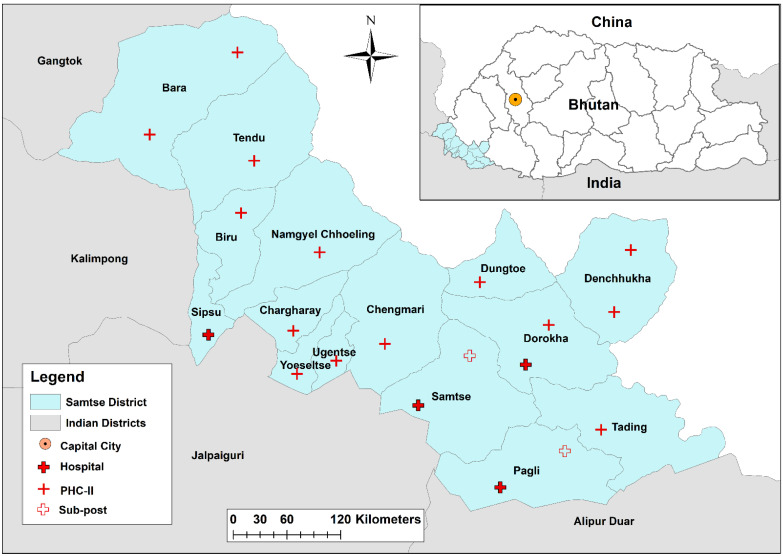
Map of Bhutan showing Samtse District and its health centres in 2021. PHC—Primary Health Centre. Note: This map is not authoritative on its international boundary.

**Figure 2 ijerph-19-07859-f002:**
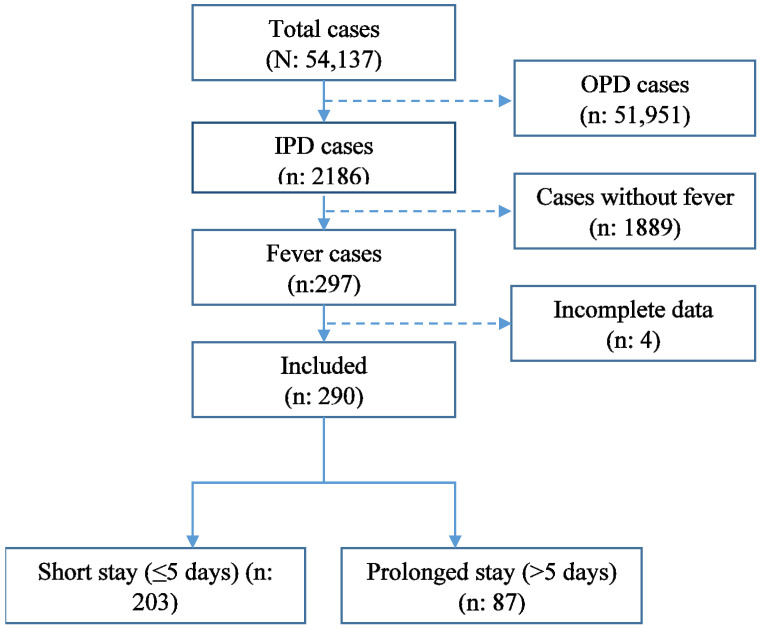
Flowchart on the assessment of hospital records of patients with fever admitted at the Samtse Hospital, Bhutan, 2020. IPD—Inpatient Department; OPD—Outpatient Department.

**Figure 3 ijerph-19-07859-f003:**
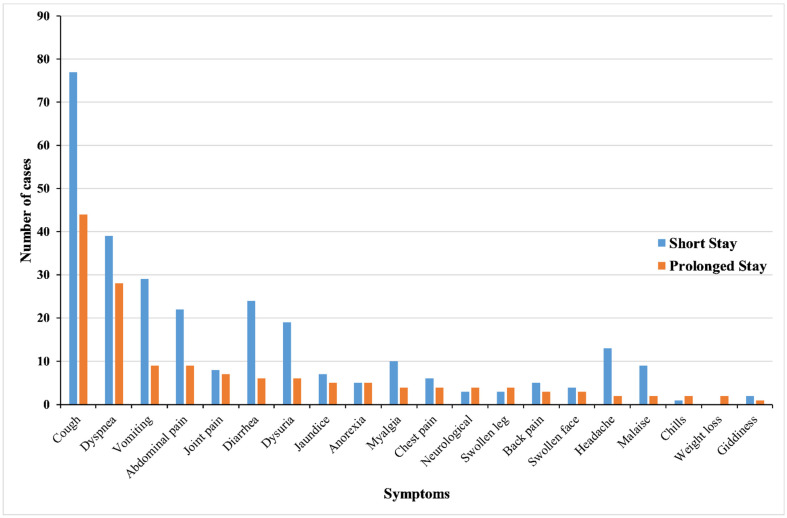
Symptoms at presentation stratified according to the length of hospital stay among patients admitted with fever in Samtse Hospital, Bhutan, 2020.

**Figure 4 ijerph-19-07859-f004:**
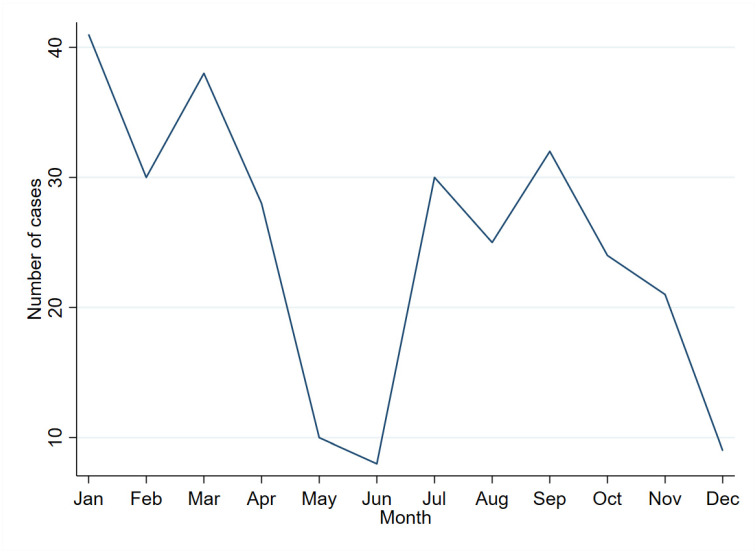
Monthly trend of the number of fever admissions at Samtse Hospital, Bhutan, 2020.

**Table 1 ijerph-19-07859-t001:** Characteristics of fever admissions by the length of stay in Samtse Hospital, Bhutan, 2020.

Category		Total	Short Stay	Prolonged Stay	*p* Value
Age (years)					
	≤12 *	135 (46.6)	90 (44.3)	45 (51.7)	0.248
	>12	155 (53.5)	113 (55.7)	42 (48.3)	
Sex					
	Male	167 (57.6)	112 (55.2)	55 (63.2)	0.204
	Female	123 (42.4)	91 (44.8)	32 (36.8)	
Occupation		(0.0)	(0.0)	(0.0)	0.379
	Children/students	151 (52.1)	102 (50.3)	49 (56.3)	
	Farmer	81 (27.9)	56 (27.6)	25 (28.7)	
	Housewife	37 (12.8)	27 (13.3)	10 (11.5)	
	Others **	21 (7.2)	18 (8.9)	3 (3.5)	
Residence					
	Urban	53 (18.3)	47 (23.2)	6 (6.9)	0.001
	Rural	237 (81.7)	156 (76.9)	81 (93.1)	
Source					
	Catchment population of Samtse hospital	266 (91.7)	187 (92.1)	79 (90.8)	0.742
	Referred from other health centres	24 (8.3)	16 (7.9)	8 (9.2)	
Outcome					
	Recovered	260 (90.0)	180 (89.1)	80 (92.0)	0.761
	Referred to higher centre	25 (8.7)	19 (9.4)	6 (6.9)	
	Dead	4 (1.4)	3 (1.5)	1 (1.2)	
Co-morbidities					
	No	165 (81.3)	69 (79.31)	234 (80.7)	0.697
	Yes ***	56 (19.3)	38 (18.7)	18 (20.7)	
	Hypertension	12 (21.4)	9 (23.7)	3 (16.7)	
	Diabetes	7 (12.5)	5 (13.2)	2 (11.1)	
	Alcohol liver disease	7 (12.5)	2 (5.3)	5 (27.8)	
	Others ****	30 (53.6)	22 (57.9)	8 (44.4)	

* Paediatric patients. ** Monk, Teacher, Armed forces personnel. *** The list of co-morbidities is provided below according to the classification of the length of hospital stay. **** Anaemia, malnutrition, seizure, stroke, sexually transmitted infections, heart and kidney diseases.

**Table 2 ijerph-19-07859-t002:** Aetiologies of fever admissions in Samtse Hospital, Bhutan, 2020.

Fever Aetiology	Total	Short Stay	Prolonged Stay
Respiratory tract infection *	85 (29.3)	51 (25.1)	34 (39.1)
Acute undifferentiated febrile illness	48 (16.6)	41 (20.2)	7 (8.1)
Acute gastroenteritis	20 (6.9)	18 (8.9)	2 (2.3)
Urinary tract infection and kidney disease	17 (5.9)	9 (4.4)	8 (9.2)
Sepsis	44 (15.2)	31 (15.3)	13 (14.9)
Scrub typhus	10 (3.5)	7 (3.5)	3 (3.5)
Others **	66 (22.8)	46 (22.7)	20 (23.0)

* included bronchiolitis, pleural effusion, tuberculosis, lower respiratory tract infection and pneumonia. ** included abdominal distension, candidiasis, dengue fever, malaria, rabies, tonsillitis, parotitis, neurological disease, typhoid fever, abscess, hepatitis and appendicitis.

**Table 3 ijerph-19-07859-t003:** Multivariable logistic regression of factors associated with the prolonged fever admissions in Samtse Hospital, Bhutan, 2020.

Category		Unadjusted Analysis			Adjusted Analysis		
		OR	95% CI	*p* Value	AOR	95% CI	*p* Value
Age (years)							
	<12	Ref			Ref		
	>12	0.74	0.45–1.23	0.25	0.62	0.18–2.15	0.45
Sex							
	Male	Ref					
	Female	0.72	0.43–1.20	0.21	0.56	0.30–1.07	0.08
Occupation							
	Children/students	Ref			Ref		
	Farmer	0.93	0.52–1.66	0.81	1.03	0.28–3.77	0.96
	Housewife	0.77	0.36–1.72	0.53	1.53	0.36–6.57	0.57
	Others	0.35	0.10–1.23	0.10	0.52	0.09–3.04	0.47
Residence							
	Urban						
	Rural	4.07	1.67–9.92	0.002	4.02	1.58–10.24	<0.001
Aetiology							
	AGE	Ref			Ref		
	Respiratory infection	6.00	1.31–27.54	0.021	5.30	1.11–25.39	0.04
	UTI and kidney disease	8.00	1.40–45.76	0.019	8.16	1.33–49.96	0.02
	AUFI	1.54	0.29–8.13	0.613	1.39	0.25–7.79	0.71
	Sepsis	3.77	0.76–18.66	0.103	4.30	0.80–22.98	0.09
	Scrub typhus	3.86	0.53–28.24	0.184	4.46	0.56–35.85	0.16
	Others	3.91	0.83–18.48	0.085	4.24	0.85–21.08	0.08

OR—odds ratio; AOR—adjusted odds ratio; CI—confidence interval; AGE—acute gastroenteritis; UTI—urinary tract infection; AUFI—acute undifferentiated febrile illness; *p*-value significant at <0.05.

## Data Availability

The data presented in this study are available on request from the corresponding author.

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
