# Peer review of "Aetiologies and Risk Factors of Prolonged Fever Admission in Samtse Hospital, Bhutan, 2020"

_ijerph, 2022, doi:10.3390/ijerph19137859_

Round 1

Reviewer 1 Report

Thank you for the opportunity to review the manuscript Aetiologies and risk factors of prolonged fever admission in 2 Samtse Hospital, Bhutan, 2020. 

Major comments:

Please clarify introduction/study purpose. It indicates studying fever presentation and then specifies prolonged fever (line 59) when I think it’s fever prompting prolonged stay? Please clarify and if indeed prolonged fever please indicate how prolonged fever was defined.

Please indicate in study population why patients with fever of unknown origin who were already inpatients were not included in the study.

 Would be interested in seeing an expanded study to include the patients w/o fever as well to do a comparison of how fever specifically impacted length of stay. Likely out of scope but something to consider or at minimum from the patients without fever that had comparable clinical syndromes to those patients with fever to evaluate how patients with fever with same diagnosis differed….?

Minor comments:

Abstract line 12: rewrite first sentence ‘--- Febrile illness is a common cause of---"

Line 54: rewrite sentence “—malaria is on the decline in recent years” Or write to have proper syntax.

Key words: prolonged itself is not a key word/medical term. Please edit or remove

Author Response

Reviewer #1

Major comments:

  1. Please clarify introduction/study purpose. It indicates studying fever presentation and then specifies prolonged fever (line 59) when I think it’s fever prompting prolonged stay? Please clarify and if indeed prolonged fever please indicate how prolonged fever was defined.

Response: We have revised as suggested by the reviewer.

Page 2, line numbers 60-62:

This study investigated the aetiologies and risk factors of prolonged fever defined as that lasting >5 days of admission due to fever in Samtse Hospital, Bhutan.

  1. Please indicate in study population why patients with fever of unknown origin who were already inpatients were not included in the study.

Response: Fever of unknown origin is included in the study with the term “acute undifferentiated febrile illness” (AUFI). This is the nomenclature routinely followed in clinical settings in Bhutan. To make readers understand that fever of unknown origin and AUFI means the same thing, this term has been added.

Page 4, line numbers 115-116:

For this study, aetiologies were categorised as: acute gastroenteritis, respiratory tract infection (bronchiolitis, pleural effusion, tuberculosis, lower respiratory tract infection and pneumonia), fever of unknown origin or acute undifferentiated febrile illness (AUFI - refers to other febrile illness not confirmed with the specific laboratory test), sepsis, scrub typhus and others.

  1. Would be interested in seeing an expanded study to include the patients w/o fever as well to do a comparison of how fever specifically impacted length of stay. Likely out of scope but something to consider or at minimum from the patients without fever that had comparable clinical syndromes to those patients with fever to evaluate how patients with fever with same diagnosis differed….?

Response: Thanks for the suggestion to include patients without fever and compare how fever impacted the length of stay. However, we don’t have access to the data on patients without fever because of administrative and ethical approval. Seeking approval for these data will lead to significant delays in publishing this paper. Nevertheless, we would have added this to be considered in the future studies.

Page 11, line numbers 297-298:

Further, it is suggested to compare prolonged admission better fever and non-fever admissions in Bhutan.

Minor comments:

  1. Abstract line 12: rewrite first sentence ‘--- Febrile illness is a common cause of---"

Response. We have revised as suggested.

Page 1, line number 12:

Febrile illness is a common cause of hospital admission in developing countries including Bhutan.

  1. Line 54: rewrite sentence “—malaria is on the decline in recent years” Or write to have proper syntax.

Response: Thanks for the suggested changes, we have revised accordingly.

Page 2, line numbers 55-56:

Other reported causes of fever admissions include typhoid, chikungunya and respiratory illness [18, 19], while the incidence of malaria is on the decline in recent years.

  1. Key words: prolonged itself is not a key word/medical term. Please edit or remove.

Response: We have changed the keywords as suggested by the reviewers.

Page 1, line numbers 28-29:

Key words: Developing countries; prolonged fever admission; infections; health services; epidemiology; aetiologies; risk factors

Reviewer 2 Report

Dear Authors

Greetings

I send you some observations and suggestions. Please read the attached pdf using Adobe.

Kind regards

Author Response

Reviewer #2

  1. I suggest:

Keywords: Developing countries; Prolonged fever admission; Infections; Health services; Epidemiology; aetiologies and risk factors.

Response: We have changed the keywords as suggested.

Page 1, line numbers 28-29:

Keywords: Developing countries; prolonged fever admission; infections; health services; epidemiology; aetiologies; risk factors.

  1. Viruses such as.

Response: We have added the lists of viruses and bacteria as suggested.

Page 1, line numbers 33-35:

Viral infections such as dengue, influenza and yellow fever, and bacterial infections, such as leptospirosis, typhoid and scrub typhus account for more than 80% of the pathogens causing febrile illness [2, 3].

  1. Intermittent, remittent, continuous or sustained, hectic or relapsing.

Response: Our medical record captures information on whether patients have fever on the day of admission but there is no classification of fever as intermittent or remittent or continuous.

  1. Did you apply some comparisons of univariate and multivariate bias-adjusting methods?

Response: We did not compare univariate and multivariate bias-adjusting methods in the study.

  1. The cumulative number of cases reported from 1993 until November 2019 is 687 (359 male and 328 female). At present, there are about 522 people living with HIV in the country.

Response: We did not make any changes based on this comment. The comment is related to HIV in the whole country and is taken as information.

  1. Type of fever …continuous, intermittent, remittent, continuous or sustained, hectic, and relapsing

Response: Our medical record captures information on whether patients have fever on the day of admission but there is no classification of fever as intermittent or remittent or continuous.

  1. The authors can provide better graphs for the results.

Response: Thanks for this suggestion. We have recreated a new graph.

  1. SARS-COV-2 and others?

Response: Thanks for bring this to the notice of authors. The “respiratory tract infection” did not include SARS-COV-2 and included other respiratory illness. Therefore, we have included the definition of respiratory tract infection others in the foot note of Table 2.

Page 8, line numbers 195-197:

*included bronchiolitis, pleural effusion, tuberculosis, lower respiratory tract infection and pneumonia

**abdominal distension, candidiasis, dengue fever, malaria, rabies, tonsillitis, parotitis, neurological disease, typhoid fever, abscess, hepatitis and appendicitis

  1. Univariate logistic regression is a model with only one dependent variable. A multivariate logistic regression is a model with more than one dependent variable.

Response: We agree that univariate logistic regression is a model with one dependent variable. Variables with p=0.2 in univariate logistic regression were included for multivariable logistic regression. The detail analysis has been explained in the methods section- page 4, line numbers 134-136. However, to make it clear we have added “after adjusting for other variables” in multivariable logistic regression in the revised manuscript.

Page 9, line number 201:

In the multivariable logistic regression, after adjusting for other variables, …

  1. Here, it is the only moment authors cite authors cite the COVID-19 pandemic……The manuscript did not give any comment for COVID-19, And the COVID-19 pandemic?????

Response: Thanks for highlighting this. However, we would like report that the first case of COVID-19 was reported on 9 May 2021 in Samtse Hospital (Karma et al. 2022) (https://pubmed.ncbi.nlm.nih.gov/35586007/). We have added this information in the revised manuscript.

Page 10, line numbers 236-237:

The first case of COVID-19 in Samtse District was reported on 9 May 2021 [29].

  1. Reference formatting and suggested reference

Response: Thanks for highlighting them to us. We have formatted it correctly in the revised manuscript. However, suggested reference were not included because in this study we do not report any COVID-19 cases.

Reviewer 3 Report

In this study, the authors reviewed cases of fever in Samtse Hospital in Bhutan and presented their demographic and clinical characteristics, and their association with prolonged hospital stay. Many where children, about a third stayed at the hospital for more than five days, and those from rural areas and those suffering from a respiratory, urinary or kidney diseases were more likely (than those with gastrointestinal infections) to be hospitalized longer.

This is an interesting and well-written paper given the lack of epidemiological surveillance data from several low and middle-income countries where febrile diseases are still debilitating.

There are, however, some issues that the authors should address to improve their manuscript:

1.     What is the background of patients visiting the hospital or the catchment area of the hospital? How do they differ from other areas of Bhutan?

2.     “Case definition”, pages 3–4. The authors should mention when fever was recorded in these patients. Was it at admission or at any time during hospital admission? Did patients receive any interventions or surgery?

3.     Table 1: Could the authors explain why age was dichotomized at 12 years?

4.     Figure 2: Does “OPD” and “IPD” stand for “outpatient” and “inpatient”, respectively. Please clarify in the figure caption.

Author Response

Reviewer #2

  1. In this study, the authors reviewed cases of fever in Samtse Hospital in Bhutan and presented their demographic and clinical characteristics and their association with prolonged hospital stay. Many were children, about a third stayed at the hospital for more than five days, and those from rural areas and those suffering from respiratory, urinary or kidney diseases were more likely (than those with gastrointestinal infections) to be hospitalized longer.

This is an interesting and well-written paper given the lack of epidemiological surveillance data from several low and middle-income countries where febrile diseases are still debilitating.

Response: We thank the reviewer for the positive feedback.

There are, however, some issues that the authors should address to improve their manuscript:

  1. What is the background of patients visiting the hospital or the catchment area of the hospital? How do they differ from other areas of Bhutan?

Response: We revised the manuscript to include the background of patients visiting Samtse Hospital, which doesn’t differ much from other districts in Bhutan.

Page 2, line numbers 79-80:

Patients' background in Samtse Hospital is not different from other district hospitals in the country.    

  1. “Case definition”, pages 3–4. The authors should mention when fever was recorded in these patients. Was it at admission or at any time during hospital admission? Did patients receive any interventions or surgery?

Response: Fever was recorded during the admission. This change has been made in the revised manuscript. We do not have information on interventions or surgery in the pro forma (supplementary Table S1).

Page 4, line numbers 108-110:

Fever was defined as at least one episode of axillary temperature ≥98.9 ℉ in the morning or ≥99.9 ℉ in the evening at the time of admission recorded through a calibrated thermometer.

  1. Table 1: Could the authors explain why age was dichotomized at 12 years?

Response: The age categorization was based on the clinical practice in Bhutan where patients aged up to 12 years receive paediatric care. Accordingly, we have updated the footnotes of the table.

Page 5, line number 147:

*Paediatric patients

  1. Figure 2: Does “OPD” and “IPD” stand for “outpatient” and “inpatient”, respectively. Please clarify in the figure caption.

Response: We have explained the terms in the figure caption to make it clear to the readers.

 Page 6, line number 153:

IPD- Inpatient Department; OPD- Outpatient Department.